# Transparent Zinc Oxide Memristor Structures: Magnetron Sputtering of Thin Films, Resistive Switching Investigation, and Crossbar Array Fabrication

**DOI:** 10.3390/nano14231901

**Published:** 2024-11-27

**Authors:** Alexander V. Saenko, Roman V. Tominov, Igor L. Jityaev, Zakhar E. Vakulov, Vadim I. Avilov, Nikita V. Polupanov, Vladimir A. Smirnov

**Affiliations:** 1Research Laboratory Neuroelectronics and Memristive Nanomaterials (NEUROMENA Lab), Institute of Nanotechnologies, Electronics and Electronic Equipment Engineering, Southern Federal University, 347922 Taganrog, Russia; avsaenko@sfedu.ru (A.V.S.); tominov@sfedu.ru (R.V.T.); izhityaev@sfedu.ru (I.L.J.); zvakulov@sfedu.ru (Z.E.V.); avilovvi@sfedu.ru (V.I.A.); npolupanov@sfedu.ru (N.V.P.); 2Department of Radioelectronics and Nanoelectronics, Institute of Nanotechnologies, Electronics and Electronic Equipment Engineering, Southern Federal University, 347922 Taganrog, Russia

**Keywords:** ZnO thin films, RF magnetron sputtering, structural properties, transparent memristor, ReRAM, resistive switching, simulation, crossbar

## Abstract

This paper presents the results of experimental studies of the influence of high-frequency magnetron sputtering power on the structural and electrophysical properties of nanocrystalline ZnO films. It is shown that at a magnetron sputtering power of 75 W in an argon atmosphere at room temperature, ZnO films have a relatively smooth surface and a uniform nanocrystalline structure. Based on the results obtained, the formation and study of resistive switching of transparent ITO/ZnO/ITO memristor structures as well as a crossbar array based on them were performed. It is demonstrated that memristor structures based on ZnO films obtained at a magnetron sputtering power of 75 W exhibit stable resistive switching for 1000 cycles between high resistance states (HRS = 537.4 ± 26.7 Ω) and low resistance states (LRS = 291.4 ± 38.5 Ω), while the resistance ratio in HRS/LRS is ~1.8. On the basis of the experimental findings, we carried out mathematical modeling of the resistive switching of this structure, and it demonstrated that the regions with an increase in the electric field strength along the edge of the upper electrode become the main sources of oxygen vacancy generation in ZnO film. A crossbar array of 16 transparent ITO/ZnO/ITO memristor structures was also fabricated, demonstrating 20,000 resistive switching cycles between LRS = 13.8 ± 1.4 kΩ and HRS = 34.8 ± 2.6 kΩ for all devices, with a resistance ratio of HRS/LRS of ~2.5. The obtained results can be used in the development of technological processes for the manufacturing of transparent memristor crossbars for neuromorphic structures of machine vision, robotics, and artificial intelligence systems.

## 1. Introduction

A memristor is a promising nonvolatile resistive memory device due to its advantages, such as high performance, long-term data storage, low power consumption, and multilevel behavior. A memristor is capable of changing its resistance depending on the voltage applied, thus imitating the role of a synapse in the nervous system. This provides wide opportunities for developing neuromorphic electronic devices, such as general-purpose memory devices (ReRAM), computing systems (neurocomputers), and biosensors and neurointerfaces for robotics [1,2,3,4,5,6,7,8,9,10,11,12,13,14,15]. The development of transparent memristors is associated with their use in optoelectronic neuromorphic devices, in particular, artificial machine vision systems, where the optical transparency of memristors and their synaptic behavior are the most important, as a memristor directly responds to optical action, has a memory, and can be used in systems processing sensory data and visual information in real time. Transparent memristors can also be integrated into transparent electronics devices such as thin-film transistors and diodes. In addition, memristors with optical biostability can find wide application in the creation of functional elements for optoelectronics (high-speed optical switches and light modulators) and displays that are integrated, for example, into car windows and windshields. To create efficient transparent memristors, optically transparent and electrically switchable oxide materials are required, as well as transparent top and bottom electrodes such as ITO [3,8,11,12,13,14,15,16,17,18,19,20,21].

Recently, zinc oxide (ZnO) has emerged as a promising memristor nanomaterial. It is non-toxic and has a suitable band gap (~3.4 eV) and a unique combination of optical and electrophysical properties, which are largely determined by the concentration of free charge carriers controlled by oxygen vacancies. ZnO also exhibits the synaptic behavior necessary for the creation of transparent memristor structures [16,17,18,19,20,21,22,23,24,25,26,27,28].

In the deposition of ZnO thin films, such methods as direct current (DC) and radio frequency (RF) magnetron sputtering, chemical vapor deposition (CVD), pulsed laser deposition (PLD), and sol-gel methods (spin coating) are widely used [2,4,22,23,24,25,26,27,28,29,30,31,32,33,34,35,36,37,38,39,40,41,42,43]. The magnetron sputtering method has such advantages as the ability to precisely control the parameters of the growing film during the deposition process, the high stability of the process over time, and the absence of heating the substrate to high temperatures. Usually, two methods for the magnetron sputtering of metal oxides are distinguished: sputtering of a metal target in a reactive gas environment (oxygen) or a ceramic target in an inert gas atmosphere (argon). In this case, the sputtering of a ceramic target is characterized by high process stability and the high quality of the resulting films, in contrast to reactive sputtering (low process stability and difficulty with regard to control), which may be promising for the creation of memristor structures [12,14].

This paper presents the results of a study of the influence of radio frequency (RF) magnetron sputtering of a ceramic target in an argon atmosphere at room temperature on the structural and electrophysical properties of ZnO films, the formation and study of the resistive switching of transparent ITO/ZnO/ITO memristor structures, and a crossbar array based on them.

## 2. Materials and Methods

Transparent ITO/ZnO/ITO memristor structures (Figure 1) were formed in three stages by magnetron sputtering using a VSE-PVD-DESK-PRO setup (AkademVak, Novosibirsk, Russia). In the first stage, 200 nm thick ITO films (bottom electrode) were deposited on glass substrates using pulsed direct current (p-DC) magnetron sputtering at a frequency of 100 kHz in an argon atmosphere at room temperature, a sputtering power of 200 W, and an operating pressure of 2 × 10^−3^ mbar. In the second stage, 60 nm thick nanocrystalline ZnO films were deposited on glass substrates with a transparent conducting ITO layer using RF magnetron sputtering of a 99.99% pure ZnO ceramic target (Kurt J. Lesker) in an atmosphere of argon at room temperature [1,3,15]. The sputtering power was varied from 25 to 100 W, and the working argon pressure was set at 5 × 10^−3^ mbar. The deposition time varied from 4 to 15 min, during which the thickness of the obtained ZnO thin films was 56 to 69 nm. In the third stage, the upper ITO electrode with a thickness of 150 nm and a diameter of 550 μm was deposited on the surface of ZnO film by a p-DC magnetron sputtering method in an atmosphere of argon at room temperature (Figure 1a,b) [13,15,39]. To evaluate the transparency of the ITO/ZnO/ITO memristor, the transmittance of the structure was measured in the range of 350–900 nm (Figure 1). The average transmittance in the visible spectrum (wavelength 400–800 nm) was approximately 86%, which corresponds to high optical transparency combined with promising memory capacity for neuromorphic structures of machine vision, robotics, and artificial intelligence systems.

The transparent 4 × 4 crossbar of the memristor structures was also manufactured in three stages. In the first stage, a transparent ITO film was applied to glass substrates using the pulsed magnetron sputtering method, on the surface of which photolithography was then performed and the lower electrodes were formed. In the second stage, a film of FP-383 photoresist (Frast-M, Zelenograd, Russia) was applied using the centrifugation method, which was exposed through a photomask with an array of memristor structure cells and developed in a 5% KOH solution. A film of nanocrystalline ZnO was then applied to the resulting structure using a RF magnetron sputtering method at a power of 75 W. Then, after lift-off lithography of the ZnO film, the memristor structure cells were obtained. In the third stage, the upper ITO contacts were formed using p-DC magnetron sputtering and lift-off lithography methods. To enable the possibility of conducting probe electrical measurements, we manufactured each electrode with a contact pad.

The surface morphology of ZnO films was studied using a Nova Nanolab 600 scanning electron microscope (FEI Company (Thermo Fisher Scientific, Waltham, MA, USA) and an atomic force microscope (AFM) in tapping mode on an NTEGRA scanning probe microscope (NT-MDT, Zelenograd, Russia). The electrophysical parameters of ZnO films, such as the resistivity, concentration, and mobility of charge carriers, were measured by the four-probe Hall effect method on an Ecopia HMS-3000 setup (Ecopia Co., Anyang, Republic of Korea). The chemical composition of ZnO films was determined by X-ray photoelectron spectroscopy (XPS) on a K-Alpha spectrometer (Thermo Scientific, Waltham, MA, USA) using monochromatic AlKα radiation (photon energy 1486.6 eV, spot size 400 μm). The transmittance of memristor structures in the ultraviolet and visible spectral range (350–900 nm) was determined using a UV-VIS Evolution-300 spectrophotometer (Thermo). The current-voltage characteristics of the memristor structures were measured at room temperature in air using a Keithley 4200-SCS semiconductor parameter measurement system (Keithley Instruments, Solon, OH, USA) and an EM-6070A submicron probing setup (Planar, Minsk, Republic of Belarus) with tungsten probes. The voltage was applied to the upper electrode of the memristor structure, and the lower electrode was grounded [10,11]. To study the electrical parameters of the manufactured crossbar, 1000 current-voltage characteristics were measured for each of the 16 structures in the bipolar sweep mode from −3 V to 3 V. A study of the dependence of HRS and LRS resistances on the number of switching cycles was also conducted. For this purpose, 20,000 switching cycles were applied to each structure at a reading voltage of 0.4 V. There are several ways to minimize the effects of leakage currents in passive crossbars [44,45,46]: (1) by applying voltages equal to U/2 V and U/3 V to unused electrodes; (2) by grounding the unused electrodes. In this study, to minimize the effect of leakage currents, all unused crossbar electrodes were grounded during electrical measurements. Moreover, it should be noted that a low HRS/LRS ratio leads to a reduced influence of leakage currents in passive crossbars.

## 3. Results and Discussion

Figure 2 shows the SEM and AFM images of the ZnO film surface at RF magnetron sputtering powers from 25 to 100 W. An analysis of the SEM images showed that ZnO films obtained at RF magnetron sputtering powers from 25 to 75 W have a uniform granular surface morphology, without significant cracks and pores, with an average grain size of 12.8 to 28.6 nm (Figure 2a–c). At the same time, ZnO films obtained at a sputtering power of 100 W (Figure 2d) have significant inhomogeneities associated with a changing grain size from 14.3 to 53.5 nm (the average grain size is 35.7 nm). An analysis of the AFM images showed that ZnO films have a relatively smooth surface at magnetron sputtering powers from 25 to 75 W. Thus, at a sputtering power of 25 W, the surface roughness of the film is 2.8 nm (Figure 2e), at a sputtering power of 50 W, it is 5.1 nm (Figure 2b), and at a sputtering power of 75 W, it is about 6.3 nm (Figure 2g). At a sputtering power of 100 W, an increase in the surface roughness to 11.4 nm is observed (Figure 2h) due to the presence of large grains (40–60 nm) in the film. Therefore, an increase in the power of RF magnetron sputtering gives the deposited atoms the necessary kinetic energy for the growth of larger grains, which leads to an increase in the surface roughness and can lead to a decrease in the stability of resistive switching [2,16,17,18,23].

Moreover, at low power (25 W), the roughness value is determined by the amorphous-like microstructure of the obtained samples: the formation of shallower cavities or channels contributes to the formation of a smooth surface with lower roughness. Increasing the power up to 100 W leads to changes in the structure of the films, the formation of deeper channels, and an increase in roughness. A further increase of power can lead to transformation of the crystal structure and consequently reduce the surface roughness of films [47]. Thus, the dependence of roughness on sputtering power is closely related to the peculiarities of the crystal structure, grain size, crystal texture, and distribution of defects along the grain boundaries. It should be noted that substrate temperature and gas flow can also be varied to control surface roughness [48,49]. Increasing the substrate temperature during magnetron sputtering is expected to result in the formation of finer-grained films; however, this approach can be used to a limited extent within the scope of this work because heating the substrate during the sputtering of the ZnO layer can degrade the parameters of the conductive ITO layer.

Figure 3 shows the ZnO film structure and the electrical properties of ZnO films at RF magnetron sputtering powers from 25 to 100 W. It is demonstrated that the cross section of all ZnO films (Figure 3a) obtained at sputtering powers from 25 to 100 W shows a columnar structure with a direction perpendicular to the substrate plane, which is often observed during the low-temperature deposition of oxide films; this is a consequence of the low mobility of the deposited particles on the substrate surface [4,18,22]. At the same time, the surface roughness of ZnO films increases with increasing RF magnetron sputtering power due to the growth of larger grains (Figure 3b). The obtained ZnO films have n-type conductivity and a relatively high specific resistance, about 10^3^–10^4^ Ω·cm. The charge carrier concentration in ZnO films changes significantly from 8.1 × 10^12^ cm^−3^ to 2.7 × 10^15^ cm^−3^, with an increase in the RF magnetron sputtering power from 25 to 75 W (Figure 3c). These changes in the charge carrier concentration can be associated with the crystallinity of ZnO films, particularly the grain size, as well as scattering at grain boundaries and defects [8,10]. The charge carrier mobility changes less significantly and has the opposite character (Figure 3d). Thus, at a magnetron sputtering power of 75 W, ZnO film exhibits the highest charge carrier concentration, determined by low energies of formation of internal non-stoichiometry defects—anion (oxygen) vacancies—which is necessary for the stable switching of memristor structures based on the filamentary mechanism.

Figure 4a–d shows the results of X-ray diffraction (XRD) analysis of the crystalline structure of ZnO films at RF magnetron sputtering powers from 25 to 100 W. The X-ray wavelength used in the study was 1.54051 Å. It was found that all ZnO films have a nanocrystalline structure with a pronounced diffraction peak at angles of 34.2°–34.4°, which corresponds to the orientation of the crystallographic plane (002) for the hexagonal structure of ZnO and the predominant direction of crystallite growth perpendicular to the substrate surface [8,10,16]. At the same time, ZnO films deposited at sputtering powers from 25 to 75 W have only the (002) peak, which indicates the absence of other crystalline phases (Figure 4a–c). An increase in sputtering power leads to an improvement in the crystallinity of ZnO films, which is confirmed by an increase in the intensity of the diffraction peaks and the grain size. At a higher sputtering power (100 W), in addition to the dominant peak (002), the appearance of additional small peaks (100) and (101) is observed, which can be explained by the probability of crystallite growth in random directions at higher deposition rates (Figure 4d) [8,16,41]. Therefore, at a high rate of RF magnetron sputtering in an argon atmosphere at room temperature, the dominant orientation of crystallites corresponding to the diffraction peak (002) will change, and the inhomogeneity of the obtained ZnO films will increase.

The chemical composition of elements in ZnO films was studied using XPS; the results are shown in Figure 4e,f. The survey X-ray photoelectron spectrum was obtained in the binding energy range of 0–1350 eV, which makes it possible to determine all the elements present on the surface of ZnO films [4,15,23,24,25,26,27,28]. High-resolution spectra of individual elements were recorded in order to ensure more accuracy in determining the positions of the peaks. An analysis of the survey spectrum (Figure 4e) showed that zinc, oxygen, and carbon are present on the surface of ZnO films. The presence of a carbon peak indicates its adsorption on the film surface from the surrounding atmosphere. The high-resolution spectrum of the Zn 2p level shows (Figure 4f) that the photoelectron peaks related to Zn 2p_3/2_ and Zn 2p_1/2_ are observed at binding energies of 1022.5 and 1045.6 eV and correspond to the oxidation state of Zn^2+^ in ZnO films. The O 1s spectrum exhibits an asymmetric peak with a shoulder at higher binding energies (Figure 4g), indicating the presence of different oxygen species. This asymmetric peak can be resolved into three Gaussian peaks at 531.0, 531.5, and 532.6 eV, which correspond to lattice oxygen (O^2−^), oxygen defects or vacancies, and oxygen contained in surface hydroxyl groups [11]. The O 1s spectrum resolution revealed that ZnO films contain oxygen vacancies, which may be responsible for the resistive switching of ITO/ZnO/ITO memristor structures [2,15,41].

To study the resistive switching of the ITO/ZnO/ITO memristor structure, we measured its electrical characteristics, developed a physical model, and performed simulations in the COMSOL Multiphysics program (Figure 5). The nanocrystalline ZnO film was deposited by RF magnetron sputtering at a power of 75 W and an argon pressure in the chamber of 5 × 10^−3^ mbar at room temperature. Figure 5a–c demonstrates the experimental current-voltage characteristic, showing the behavior of bipolar resistive switching, the dependence of the resistance on the number of switching cycles, and the cumulative probability. The current-voltage characteristic of the memristor structure was measured by changing the voltage applied to the upper electrode in the sequence 0 V → 2 V → 0 V → −2 V → 0 V, which is shown as arrows in Figure 5a [7,13,43,50,51,52,53,54]. Figure 5b shows the dependence of high (HRS) and low (LRS) resistances of ZnO film on the number of switching cycles. With an increase in the applied voltage from 0 V to 2 V, the ITO/ZnO/ITO memristor structure gradually switches from HRS to LRS, and the setup process is completed at 2 V (V_SET_). Subsequently, the memristor structure will maintain the LRS until the applied voltage decreases to −2 V (V_RESET_). Consequently, at VRESET, the reset process occurs, and the memristor switches from LRS to HRS, implementing bipolar resistive switching. Thus, the change in resistance from HRS to LRS is shown, which occurs at 2 ± 0.1 V, and from LRS to HRS at −2 ± 0.1 V, while the switching current is about 5 mA. It was found that for 1000 switching cycles, HRS and LRS are 537.4 ± 26.7 Ω and 291.4 ± 38.5 Ω, respectively. The resistance ratio in HRS/LRS is ~1.8 at a reading voltage of 0.4 V (Figure 5c). In general, the ZnO film is a dielectric, and for it to begin to exhibit the effect of resistive switching, it is necessary to create a certain number of oxygen vacancies in its volume. This can be achieved in several ways: by electroforming, by selecting sputtering modes to create films with a certain stoichiometry, and by using annealing after sputtering [17,55]. In this study, using a magnetron sputtering setup, we found the values of the chamber pressure and sputtering power parameters at which ZnO films begin to exhibit resistive switching. Thus, the presence of resistive switching in the obtained films is related to the sputtering conditions.

Based on the experimental studies of ITO/ZnO/ITO memristor structures, a physical model of the memristor was developed, the general view of which is presented in Figure 5d, which shows the case of the potential distribution in the structure at a potential of the upper electrode of 2 V. The model allows one to estimate the processes of generation and recombination of oxygen vacancies in ZnO film, as well as the subsequent change in the concentrations of vacancies and oxygen ions due to their migration under the action of the applied electric field. The model considers the difference in the surface roughness and thickness of ZnO films obtained at RF magnetron sputtering powers from 25 to 100 W. Taking into account the local nanometer geometry allows one to estimate its effect on the inhomogeneity of the distribution of the electric field strength both at the surface of the electrodes and between the electrodes in ZnO film.

Figure 5e,f shows the distributions of the electric field strength in ZnO film. It was found that the local nanometer roughness of the surface leads to non-uniformity of the electric field strength in the region of the upper electrode. In the region of the edge of the upper electrode, the greatest increase in the electric field strength was found compared to the values under the upper electrode. Figure 5g,h shows the distributions of the concentration of oxygen vacancies in ZnO film (red color corresponds to the highest concentration of vacancies, blue—to the lowest). From the obtained results, it is evident that the areas with an increase in the electric field strength become the main sources of vacancy generation in ZnO film. In this case, the highest rate of vacancy generation occurs in the region of the edge of the upper electrode. These same areas subsequently localize oxygen ions and form a vacancy-depleted region along the perimeter of the upper electrode.

Based on the results of mathematical modeling of the distribution of oxygen vacancies in ZnO film, changing its conductivity, theoretical current-voltage characteristics were constructed at different RF magnetron sputtering powers of ZnO film (Figure 5i). These findings correlate with the experimental measurements of the obtained memristor structures. At the same time, for ZnO film deposited at a RF magnetron sputtering power of 75 W, slightly higher currents were obtained, and a large difference in HRS and LRS resistances in the negative voltage region was revealed. This can be associated mainly with the nanocrystalline structure and the highest concentration of oxygen vacancies in ZnO film deposited at an RF magnetron sputtering power of 75 W.

For practical application of the transparent memristors studied in this work, and in particular, for emulation of the biological synapses on this basis as used in neuromorphic machine vision systems, it is necessary to develop and manufacture an arrays of elements that will be electrically connected to each other. It is believed that for the physical implementation of biological synapses, memristor structures manufactured according to the crossbar architecture are the most suitable ones, since they are capable of emulating computational primitives of neuronal cells [26,27,28,29,30,31,32,33,34,35,36,37,38,39,40]. Based on the above, a transparent crossbar was developed and manufactured from 16 ITO/ZnO/ITO memristor structures, with a size of 2000 × 2000 nm each (Figure 6a,e).

An analysis of the obtained current-voltage characteristics showed that the manufactured memristor structures have stable resistive switching (Figure 6b,c). The current in the HRS is about 15 μA at 0.4 V, and in a low resistance state (LRS), 32 μA. Similar results were obtained for the remaining memristor structures of the crossbar (Figure 6f). Endurance testing showed that after 20,000 switching cycles, the memristor structures switch between LRS = 16.7 ± 1.9 kΩ and HRS = 33.7 ± 1.6 kΩ for one device (Figure 6c), and between LRS = 13.8 ± 1.4 kΩ and HRS = 34.8 ± 2.6 kΩ for all devices (Figure 6g), while the resistance ratio HRS/LRS is ~2.1 and 2.5, respectively (Figure 6d,h). This difference can be explained by the influence of the surface relief at the same roughness and thickness of ZnO film on the resistive switching effect, which leads to fluctuations in the concentration and distribution profile of defects in the volume of ZnO nanocrystalline film for each ITO/ZnO/ITO memristor structure. The fabricated crossbar of ITO/ZnO/ITO-based memristor structures can be used in promising applications for robotics and artificial intelligence systems, particularly in visual systems, as well as in pattern recognition systems.

## 4. Conclusions

As a result of the conducted studies, thin ZnO films were obtained by the RF magnetron sputtering method in an argon atmosphere at room temperature. The effect of magnetron sputtering power on the structural and electrophysical properties of ZnO films was investigated. It was shown that an increase in the RF magnetron sputtering power from 25 to 100 W leads to an increase in the grain size from 12.8 to 35.7 nm and surface roughness of ZnO films from 2.8 to 11.4 nm. It was found that at a sputtering power of 75 W, ZnO films have a sufficiently smooth surface and a uniform nanocrystalline structure with the highest concentration of oxygen vacancies, which is promising for the formation of memristor structures exhibiting stable resistive switching.

Thus, the possibility of creating transparent ITO/ZnO/ITO memristor structures and a crossbar array based on these on glass substrates for use in artificial machine vision systems and non-volatile resistive memory devices is demonstrated. It is shown that the memristor structures based on ZnO films obtained at an RF magnetron sputtering power of 75 W exhibit stable resistive switching. It is demonstrated that in 1000 cycles the memristor structure switches between HRS = 537.4 ± 26.7 Ω and LRS = 291.4 ± 38.5 Ω, while the resistance ratio in the HRS/LRS is ~1.8. Mathematical modeling of transparent memristor structures based on nanocrystalline ZnO films is carried out, and its results correlate well with the obtained experimental data. It is shown that regions with increased electric field strength along the edge of the upper electrode become the main sources of the generation of oxygen vacancies in ZnO film.

A transparent crossbar of 16 ITO/ZnO/ITO memristor structures with a size of 2000 × 2000 nm each was also developed and manufactured. It was demonstrated that for 20,000 cycles, the resistive switching between LRS = 13.8 ± 1.4 kΩ and HRS = 34.8 ± 2.6 kΩ is achieved for all devices, while the resistance ratio of HRS/LRS is ~2.5. The results obtained can be used in the development of technological processes for the manufacturing of transparent memristor crossbars for neuromorphic structures of machine vision, robotics, and artificial intelligence systems.

## Figures and Tables

**Figure 1 nanomaterials-14-01901-f001:**
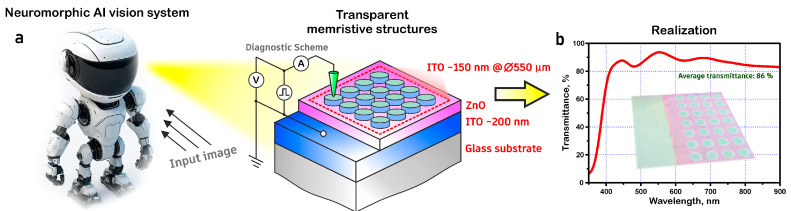
Experimental transparent ITO/ZnO/ITO memristor on glass substrate: (**a**) schematic structure; (**b**) appearance and transmission spectrum.

**Figure 2 nanomaterials-14-01901-f002:**
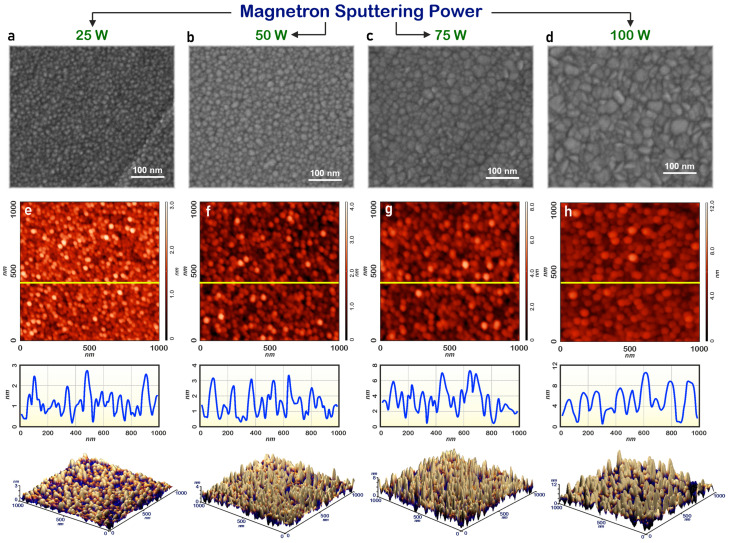
SEM and AFM images of the surface of ZnO nanocrystalline films obtained at RF magnetron sputtering power: (**a**,**e**) 25 W; (**b**,**f**) 50 W; (**c**,**g**) 75 W; (**d**,**h**) 100 W.

**Figure 3 nanomaterials-14-01901-f003:**
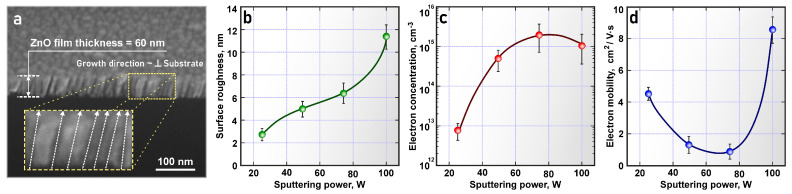
Structure of ZnO films and electrophysical properties of ZnO films: (**a**) transverse cleavage with a thickness of about 60 nm; (**b**) dependence of surface roughness on sputtering power; (**c**) dependence of charge carrier concentration on sputtering power; (**d**) charge carrier mobility on sputtering power.

**Figure 4 nanomaterials-14-01901-f004:**
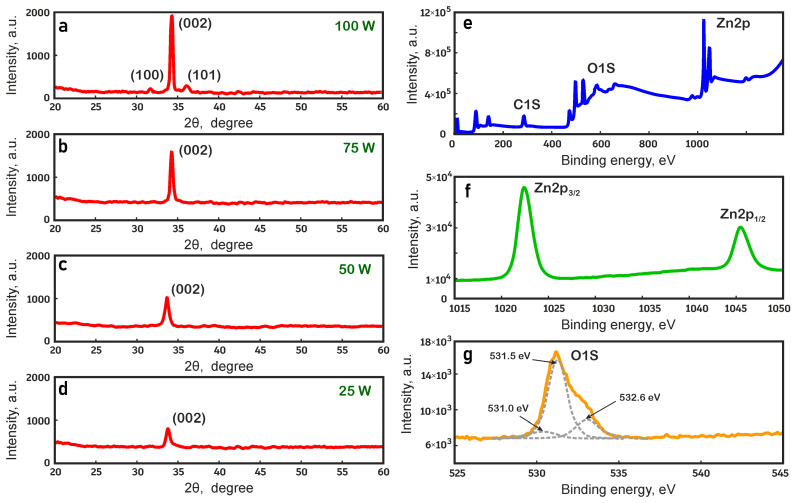
Structural properties of ZnO films obtained at different RF magnetron sputtering powers: (**a**) 25 W; (**b**) 50 W; (**c**) 75 W; (**d**) 100 W; (**e**) overview XPS spectrum of the film at 75 W; (**f**) high-resolution XPS spectrum of the zinc level; (**g**) high-resolution XPS spectrum of the oxygen level.

**Figure 5 nanomaterials-14-01901-f005:**
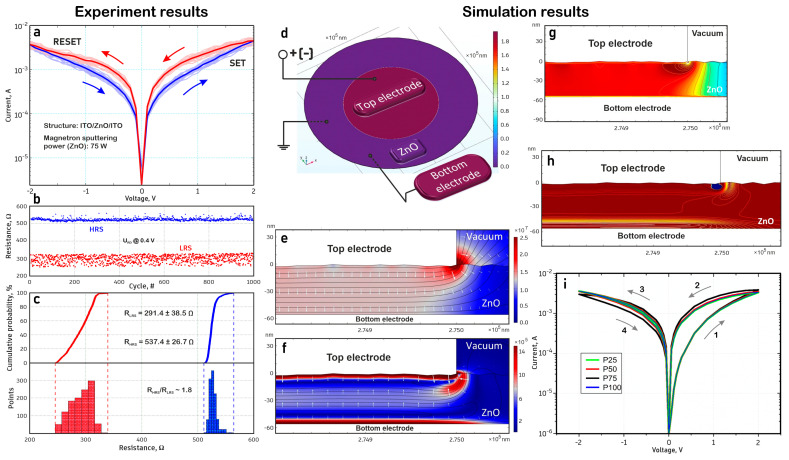
Investigation of resistive switching and modeling of transparent ITO/ZnO/ITO memristor: (**a**) experimental current-voltage characteristic; (**b**) dependence of resistance on the number of switching cycles; (**c**) cumulative probability; (**d**) general view of the memristor structure model; (**e**) initial distribution of electric field strength with equipotential lines in the upper electrode region; (**f**) distribution of electric field strength with equipotential lines in the upper electrode region, taking into account the generation/recombination and migration of vacancies; (**g**) initial distribution of vacancy concentration in the upper electrode region; (**h**) distribution of vacancy concentration in the upper electrode region, taking into account their generation/recombination and migration; (**i**) theoretical current-voltage characteristics of the memristor structure based on ZnO film obtained at different powers of RF magnetron sputtering.

**Figure 6 nanomaterials-14-01901-f006:**
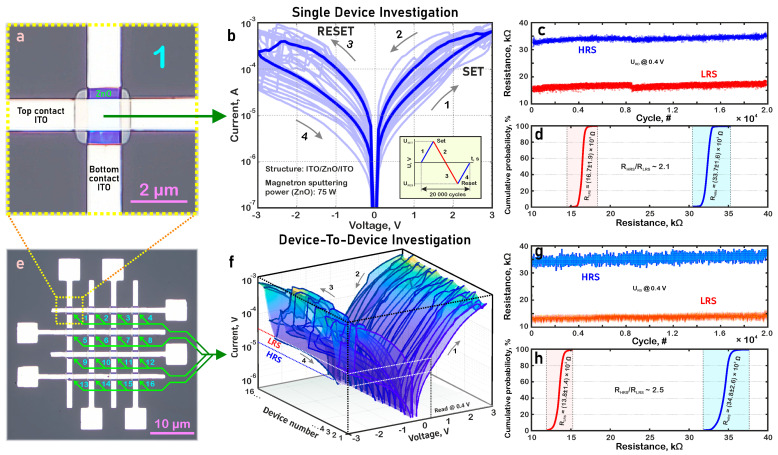
Study of resistive switching of crossbar array of 16 transparent memristor structures: (**a**) optical image of one memristor structure; (**b**) current-voltage characteristic for one memristor structure; (**c**) dependence of resistance on the number of switching cycles for one memristor structure; (**d**) cumulative probability for one memristor structure; (**e**) optical image of crossbar; (**f**) current-voltage characteristics for crossbar array; (**g**) average statistical dependence of resistance on the number of switching cycles for crossbar array; (**h**) average statistical cumulative probability for crossbar array.

## Data Availability

The original contributions presented in this study are included in the article. Further inquiries can be directed to the corresponding author.

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
