# Peer review of "Transparent Zinc Oxide Memristor Structures: Magnetron Sputtering of Thin Films, Resistive Switching Investigation, and Crossbar Array Fabrication"

_nanomaterials, 2024, doi:10.3390/nano14231901_

Round 1

Reviewer 1 Report

Comments and Suggestions for Authors

This study presents a significant contribution to the field of transparent electronics and neuromorphic computing by examining the impact of high-frequency magnetron sputtering on the structural and electrical properties of ZnO-based memristor devices. Through a methodical approach, the authors demonstrate a clear correlation between sputtering power and the film’s resistive switching performance, offering insights into the role of oxygen vacancies in resistive states. The work is scientifically sound, with rigorous experimental validation, and provides an innovative basis for further exploration in transparent crossbar arrays for neuromorphic applications, highlighting its potential for advancing machine vision and AI systems. The following issues need to be addressed before the article can be published:

1.     Optoelectronic memristors and other devices that integrate sensing and information storage are highly significant, as noted by the reviewer, who recommends adding references to this section of the manuscript, such as Adv. Mater. 2024, 36, 2405145, Chem. Rev. 2020, 120, 3941.

2.     Can the surface roughness of the ZnO films be further reduced, especially at higher sputtering powers? Discuss whether adjustments in sputtering parameters, such as lowering deposition temperature or modifying gas flow, could achieve smoother film surfaces while maintaining optimal film properties.

3.     In most memristor devices, an initial high-voltage forming process is required to create conductive pathways and enable resistive switching. Could the authors clarify why the ITO/ZnO/ITO memristor in this study does not require such a forming step? Is this due to specific properties of the ZnO film or the sputtering conditions used?

4.     Have the authors attempted using other asymmetric electrode structures, and could they predict how such a configuration might impact the performance of the memristor?

5.     Have the authors evaluated crosstalk effects in the 16-element ITO/ZnO/ITO memristor crossbar array?

Overall, the authors' work is systematic and innovative. With the resolution of the above-mentioned issues, I recommend accepting the manuscript.

Reviewer 2 Report

Comments and Suggestions for Authors

This manuscript offers a detailed and valuable exploration of how high-frequency magnetron sputtering power affects ZnO films, providing insights into their structural and electrical properties under specific conditions. The findings suggest that these ZnO-based devices could be useful for AI and robotics, supporting the development of transparent memory devices for advanced technologies.

However, several issues require clarification:

1) What is the transparency efficiency of the devices? Could the authors provide an optical image to clearly show transparency or provide experimental data on this? For instance, previous studies, such as those on ITO/LaAlO₃/SrTiO₃ devices ( ACS Appl. Mater. Interfaces 2014, 6, 8575−857), demonstrated high transparency efficiency and should be discussed in the introduction.

2) The manuscript mentions that the "O 1s spectrum asymmetric peak can be resolved into three Gaussian peaks." It would be helpful if the fitting peaks were shown to aid the reader's understanding.

3) The manuscript reports a resistance ratio of ~1.8 to ~2.5 in high and low resistance states, which is relatively low. Could the authors compare this result with previous studies? What might explain the low ratio, and are there ways to improve it? Additionally, is there significant current leakage? The authors should address these questions.

Round 2

Reviewer 1 Report

Comments and Suggestions for Authors

agree to accept the manuscript in the current version

Reviewer 2 Report

Comments and Suggestions for Authors

The authors have addressed all the questions and the paper can be acceted now.